# Visual landmarks sharpen grid cell metric and confer context specificity to neurons of the medial entorhinal cortex

**José Antonio Pérez-Escobar[1,2], Olga Kornienko[1,2], Patrick Latuske[1,2], Laura Kohler[1,2], Kevin Allen[1,2]\***

[1]Department of Clinical Neurobiology, Medical Faculty of Heidelberg University, Heidelberg, Germany; [2]German Cancer Research Center, Heidelberg, Germany

**Abstract** Neurons of the medial entorhinal cortex (MEC) provide spatial representations critical for navigation. In this network, the periodic firing fields of grid cells act as a metric element for position. The location of the grid firing fields depends on interactions between self-motion information, geometrical properties of the environment and nonmetric contextual cues. Here, we test whether visual information, including nonmetric contextual cues, also regulates the firing rate of MEC neurons. Removal of visual landmarks caused a profound impairment in grid cell periodicity. Moreover, the speed code of MEC neurons changed in darkness and the activity of border cells became less confined to environmental boundaries. Half of the MEC neurons changed their firing rate in darkness. Manipulations of nonmetric visual cues that left the boundaries of a 1D environment in place caused rate changes in grid cells. These findings reveal context specificity in the rate code of MEC neurons.

**\*For correspondence:** allen@uni-heidelberg.de

**Competing interests:** The authors declare that no competing interests exist.

## Introduction

Grid cells of the medial entorhinal cortex (MEC) fire at several locations organized as a grid of equilateral triangles (*Hafting et al., 2005*). The repetitive nature of the firing fields suggests that grid cells are an important metric element of a neuronal system involved in spatial navigation (*Hafting et al., 2005*; *McNaughton et al., 2006*; *Burgess et al., 2007*). To keep track of the animal's position during locomotion, grid cells appear to rely on self-motion cues (*Hafting et al., 2005*). Indeed, computational models suggest that grid cells integrate heading direction and traveled distance (*Fuhs, 2006*; *McNaughton et al., 2006*; *Burgess et al., 2007*; *Burak and Fiete, 2009*). The directional component likely depends on inputs from head-direction cells found in several subcortical and cortical areas, including the MEC (*Taube et al., 1990*; *Valerio and Taube, 2012*; *Winter et al., 2015*). Traveled distance, on the other hand, could be estimated from the firing rate of speed cells located in the MEC (*Kropff et al., 2015*). As expected from an internal speedometer, the increase of the firing rate of speed cells as a function of running velocity is preserved across different environments (*Kropff et al., 2015*).

Without corrections based on external inputs, intrinsic noise in neuronal networks causes an accumulation of error in the internal estimation of position (*Burgess et al., 2007*; *Burak and Fiete, 2009*). A mechanism is therefore needed to prevent the position estimate from rapidly deviating from the actual location of the animal. External landmarks, including the borders of the enclosure, serve this purpose (*Knierim et al., 1995*; *Skaggs et al., 1995*; *Evans et al., 2015*; *Hardcastle et al., 2015*). The influence of landmarks on grid cell activity is illustrated by cue control experiments in which the rotation of a single polarizing visual landmark causes an equivalent reorientation of the grid pattern (*Hafting et al., 2005*), and by the reduced spatial selectivity of grid cells in darkness

(*Hafting et al., 2005*; *Allen et al., 2014*). Moreover, compression of a familiar rectangular environment along one axis can lead to a contraction of the grid pattern in the same axis (*Barry et al., 2007*; *Stensola et al., 2012*) and grid orientation aligns to the environment borders with a small but constant offset (*Krupic et al., 2015*; *Stensola et al., 2015*). Finally, there is also evidence that the color and odor of an enclosure can affect the positioning of the grid fields but not the grid orientation or the firing rate of the neurons (*Marozzi et al., 2015*). Thus, visual landmarks within an environment prevent error accumulation in grid cell networks and anchor the grid fields to the environment.

Whether visual landmarks influence the activity of MEC neurons beyond anchoring the firing fields of grid cells is not known. On one hand, the role of visual information could be limited to ensuring long-term stability of the MEC spatial representations. In this scenario, the firing rate of MEC neurons would be context-invariant and remain unchanged by manipulations of visual landmarks (*Fyhn et al., 2007*; *Kropff et al., 2015*; *Marozzi et al., 2015*). Alternatively, visual information could have a broader impact on the MEC network. For example, the firing rate and speed code of MEC neurons could be affected by visual inputs. In addition, nonmetric contextual information, which is independent from the shape and orientation of an enclosure, could be encoded in the firing activity of MEC neurons. We set out to test these two alternative views by recording the activity of MEC neurons in mice running in environments where access to visual landmarks was manipulated.

## Results

We first performed recordings from the MEC in mice exploring an elevated circular arena (*Figure 1A*, data available from the Dryad Digital Repository: 10.5061/dryad.c261c; *Pérez-Escobar et al., 2016*). The only potential sources of visible light were 4 identical LED panels located around the arena. Recording sessions included a sequence of 60 2-min trials, alternating between light and dark trials (*Figure 1B*). On most recording sessions, 2 different light panels were used (l1 and l2) with a 90° or 180° angle between them. Dark trials following l1 and l2 trials were referred to as d1 and d2 trials, respectively. Histological examination showed that most recording sites were in the MEC (75.51%, 37 out of 49) or at the transition between the MEC and the parasubiculum (12.24%, 6 out of 49) (*Figure 1C*, *Figure 1—source data 1*, *Figure 1—source data 2*). Another 12.24% (6 out of 49) of the recording sites were in the parasubiculum. Of the tetrode tracks in the MEC, 86.49% had reached layer II by the last recording session. A total of 880 neurons were recorded in 8 mice (89 recording sessions).

### Visual landmarks control the orientation of MEC spatial representations

We first tested whether the orientation of MEC spatial representations was controlled by the position of the light source (*Muller and Kubie, 1987*; *Goodridge and Taube, 1995*; *Hafting et al., 2005*). We found that the firing fields of grid cells, border cells and irregular spatially selective neurons rotated around the center of the arena to follow the position of the light (*Figure 1D*, *Figure 1—figure supplement 1*). Correlations between maps of l1 and l2 trials were calculated after rotating l2 maps (*Figure 1E*). Peak correlations were obtained near 90° and 180° for recording sessions in which the angle between l1 and l2 was 90° and 180°, respectively. Thus, during light trials, the light panels acted as dominant polarization cues.

### Rapid degradation of grid cell periodicity in absence of visual landmarks

We investigated whether grid cells maintained a stable grid firing pattern in darkness. Surprisingly, the periodic firing was not visible in most rate maps of dark trials (*Figure 2A*). Grid scores and information scores were much lower during d1 trials compared to l1 trials (*Figure 2B*; paired Wilcoxon signed rank test, l1 vs d1, $n = 139$ grid cells, grid score: $v = 9223$, $p<10^{-16}$, information score: $v = 9722$, $p<10^{-16}$). The reductions in grid periodicity and spatial information content were also significant when comparing the medians of individual mice in which at least 5 grid cells were recorded (*Figure 2C*; paired Wilcoxon signed rank test, $n = 6$ mice, grid score: $v = 21$, $p=0.031$, information score: $v = 21$, $p=0.031$). Moreover, these alterations remained significant when limiting the analysis to neurons recorded from hemispheres in which all tetrode tips were located in the MEC (referred to as MEC tetrodes) (paired Wilcoxon signed rank test, $n = 75$ grid cells, grid score: $v = 2708$, $p<10^{-11}$,

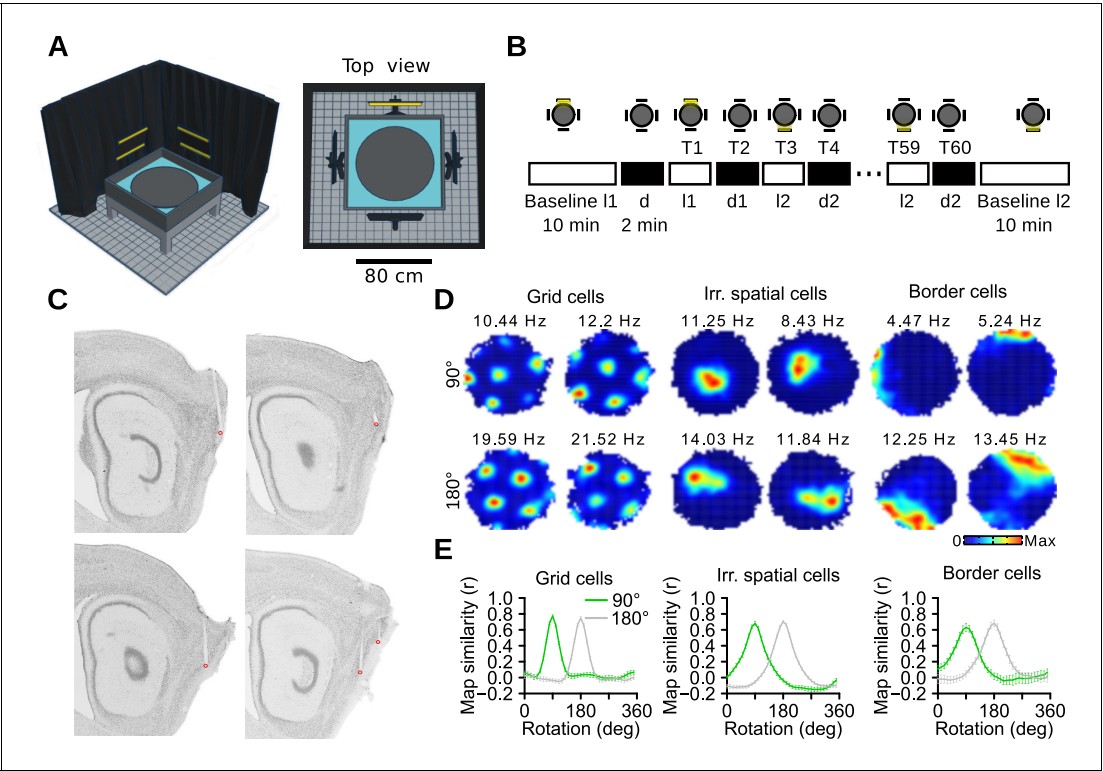

**Figure 1.** Recording protocol and cue control of MEC spatial representations. (**A**) Schematic of the recording apparatus. Left: Side perspective. An elevated circular arena was located within a square box filled with water. Note that half of the black curtain and two light panels were omitted for better visualization. Right: Top view. Four LED panels were positioned outside the box at 90° to each other. (**B**) Recording protocol. Two lights out of four were selected at the beginning of each recording session (l1 and l2). The protocol started and ended with a 10-min baseline, one with each light (Baseline l1 and Baseline l2). In between were 60 2-min trials (T), alternating between light and dark trials. The presentation of the two lights (l1 and l2) followed a random sequence. Only one light could be switched on at any time. (**C**) Sagittal brain sections showing the location of the recording sites (red dots) in the MEC. (**D**) Examples of firing rate maps of two grid cells (left, one cell on each row), two irregular spatially selective neurons (middle) and two border cells (right) during trials with l1 and l2. Top and bottom rows contained cells recorded with l1 and l2 being at 90° and 180° to each other, respectively. The numbers above the firing rate maps are the peak firing rates. (**E**) Correlations between l1 and l2 maps after rotating l2 maps in 10° steps, plotted separately for sessions with 90° and 180° between l1 and l2.

The following source data and figure supplement are available for figure 1:

**Source data 1.** Histological results of the mice recorded on the circular arena.
**Source data 2.** Location of the tetrode tips in each hemisphere for recordings done on the circular arena.
**Figure supplement 1.** Spatial scores used to identify spatially selective neurons.

information score: $v$ = 2846, p<$10^{-14}$). Thus, visual landmarks were required to stabilize the grid firing pattern.

The grid pattern observed during light trials was present during the first few seconds in darkness. Trials were divided into 12 10-s blocks. The block firing maps were correlated to the complete l1 maps to obtain a measure of map similarity relative to l1 trials (*Figure 2D*). During d1 trials, map similarity was initially high and decreased during the first 30 s (paired Wilcoxon signed rank test, *n* = 132, b1-b2: p<$10^{-16}$, b2-b3: p=0.00053). Block 3 was not different from block 4 (p=0.17). This indicates that the stable periodic pattern was only present during the first 30 s in darkness. Similar conclusions were reached when using the spike distance metric (*Hardcastle et al., 2015*) to quantify the accumulation of error in grid cell spikes during d1 trials (*Figure 2—figure supplement 1*). Interestingly, when the l1 LED panel was turned back on, map similarity during the first 10 s was lower than

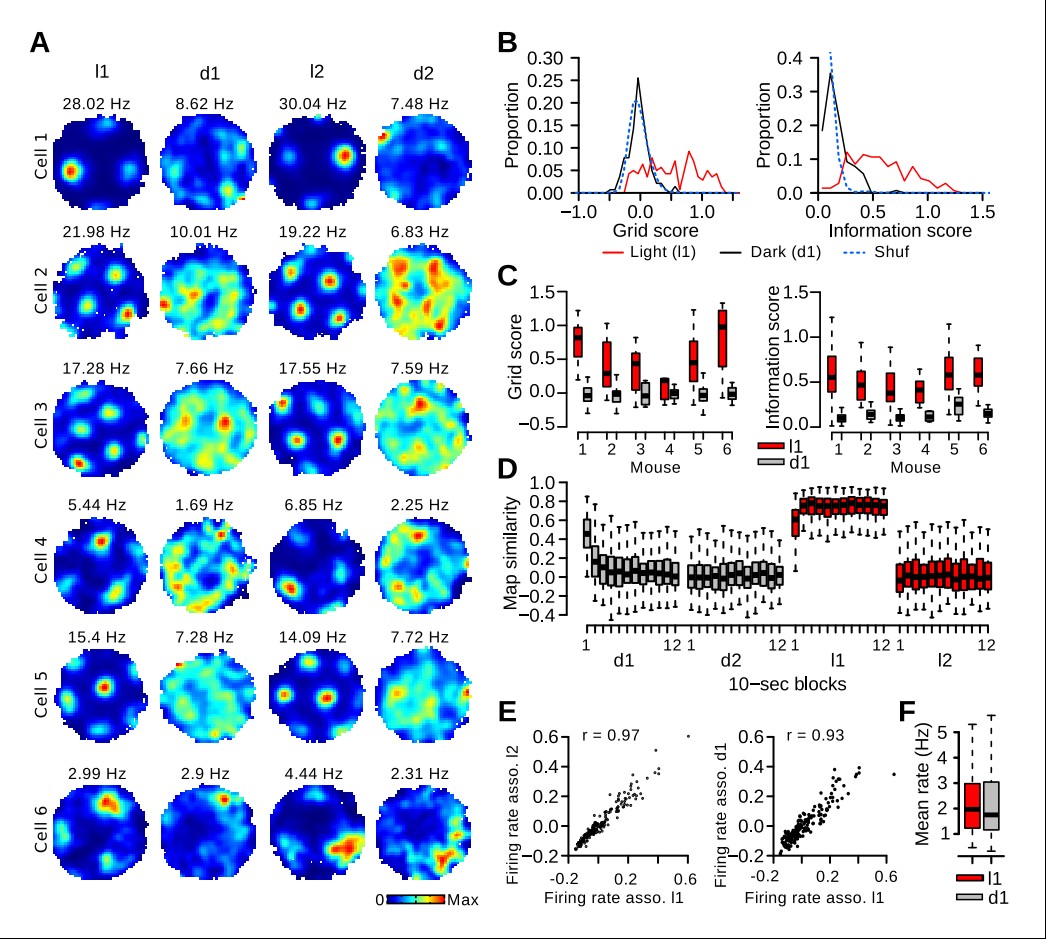

**Figure 2.** Rapid degradation of grid cell periodicity in absence of visual landmarks. (**A**) Firing maps of 6 grid cells during light and dark trials. (**B**) Distribution of grid and information scores of grid cells during l1 and d1 trials. The dotted blue line represents the surrogate (Shuf) distribution. (**C**) Grid and information scores during l1 and d1 trials for individual mice with at least 5 recorded grid cells. (**D**) Map similarity between 10-s block maps and l1 maps (left column in panel **A**). (**E**) Left: Firing rate associations of pairs of grid cells during l1 and l2 trials. Right: Firing rate associations of pairs of grid cells during l1 and d1 trials. (**F**) Mean firing rate of grid cells.

The following figure supplement is available for figure 2:

**Figure supplement 1.** Spike distance metric (SDM) during light and dark trials.

during the subsequent l1 block (*Figure 2D*, $p < 10^{-15}$). Thus, the reinstatement of the grid pattern by visual stimuli occurred over the course of several seconds.

Pairs of grid cells with a similar spacing exhibit very stable firing associations; cells that fire together in one environment also fire together in a different environment (*Fyhn et al., 2007*; *Yoon et al., 2013*; *Allen et al., 2014*). We therefore tested if the firing associations between grid cells were maintained even when the grid patterns were not stable. The instantaneous firing rate of grid cells was calculated (time window: 100 ms, gaussian smoothing kernel s.d: 100 ms) and the correlation coefficients between instantaneous firing rate vectors served as a measure of firing rate association. As expected, firing rate associations for pairs of grid cells were stable from l1 to l2 trials (*Figure 2E*, Pearson correlation of firing rate associations in l1 and l2 trials, $n = 186$ grid cell pairs, $r = 0.968$, $p < 10^{-16}$). Despite instability in the spatial firing patterns of grid cells during dark trials, the firing rate associations of grid cells were to a large extent preserved during d1 trials (*Figure 2E*, $r = 0.926$, $p < 10^{-16}$). The median firing rate of grid cells was slightly lower during dark trials than during light trials (*Figure 2F*; paired Wilcoxon signed rank test $v = 6048$, $p = 0.0129$).

## Partly preserved distance coding by grid cells in darkness

The loss of grid periodicity in darkness could be due to a slow translational drift of the grid pattern relative to the recording environment. If this is the case, the grid pattern should be visible in spike-triggered rate maps in which the effect of slow translational drift is minimized by resetting the position data each time a cell fires a spike (*Figure 3A*) (*Bonnevie et al., 2013*; *Allen et al., 2014*). In these maps, each spike in turn served as a reference spike and the position of the animal in the 10 s following a reference spike was shifted so that the position of the animal at the time of the reference spike was (0,0). Spike-triggered maps of grid cells during light trials often showed a central field surrounded by 6 fields (*Figure 3A*). During dark trials, the surrounding fields were less distinct or sometimes appeared as a ring of elevated activity. Accordingly, grid scores calculated from the spike-triggered maps were lower during d1 than l1 trials (*n* = 139 grid cells, median l1: 0.054, median d1: −0.062, *v* = 12246, p=0.00011). Thus, the impairment in grid periodicity in darkness cannot be fully explained by a slow translational drift.

We next tested whether the modulation of firing rate as a function of distance was preserved in darkness. Distance coding by grid cells was visualized by plotting distance tuning curves, i.e. the mean firing rate of a grid cell as a function of the distance from reference spikes (*Figure 3A*). To perform population analysis, distance was normalized to the grid spacing measured during the first baseline. The increase in firing rate at the first period of the grid cells (*Figure 3B*; normalized distance = 1) was quantified with a distance score, which was defined as $(\lambda_1 - \lambda_{0.5})/(\lambda_1 + \lambda_{0.5})$, where $\lambda_{0.5}$ and $\lambda_1$ represent the firing rate of a neuron at normalized distance 0.5 and 1, respectively. Distance scores were smaller during dark trials than during light trials (*Figure 3C*; paired Wilcoxon signed rank test, *n* = 50 grid cells, *v* = 1181, p<10$^{-7}$), but distance scores in darkness were still above chance levels (*v* = 1006, p=0.00038). Taken together, the results indicate that estimation of distance by grid cells over short periods (10 s) was partially preserved in darkness. Moreover, the ring of activity in the spike-triggered maps of some grid cells in darkness suggests that the orientation of the grid pattern was not stable.

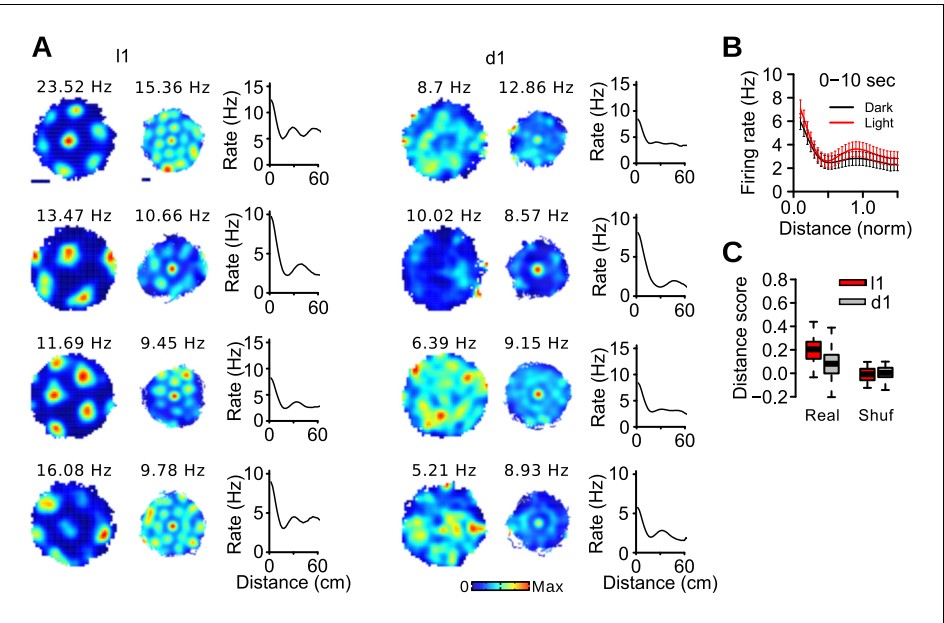

**Figure 3.** Distance coding in grid cells. (**A**) Examples of firing rate maps, spike-triggered firing maps and distance tuning curves of 4 grid cells during light and dark trials. The scale bars for firing maps represent 20 cm. (**B**) Mean distance tuning curves of grid cells. Distance was normalized to the spacing of each grid cell. (**C**) Distance score of grid cells during l1 and d1 trials for real and surrogate (Shuf) data.

## Influence of visual information on speed coding in the MEC

It has been suggested that a population of entorhinal neurons provides a context-invariant running speed estimate (*Kropff et al., 2015*). We tested whether the speed-rate function of speed-modulated MEC cells was preserved when visual inputs were absent. The correlation coefficients between instantaneous firing rate vectors and running speed served as speed scores to identify cells with significant speed modulation. Examples of neurons with significant speed modulation are shown on *Figure 4A*. Out of 880 MEC neurons, 304 (34.55%) had a speed score above chance levels (*Figure 4B*). Speed scores were negatively correlated with spatial information scores (Pearson correlation, $n = 880$, $r = -0.146$, $p<10^{-5}$), but several spatially selective neurons had significant speed scores (grid cells: 67 out of 139, irregular spatially selective cells: 54 out of 226, border cells: 11 out of 63, head-direction cells: 30 out of 85). Speed scores were positively correlated with mean firing rates ($n = 880$, $r = 0.238$, $p<10^{-13}$).

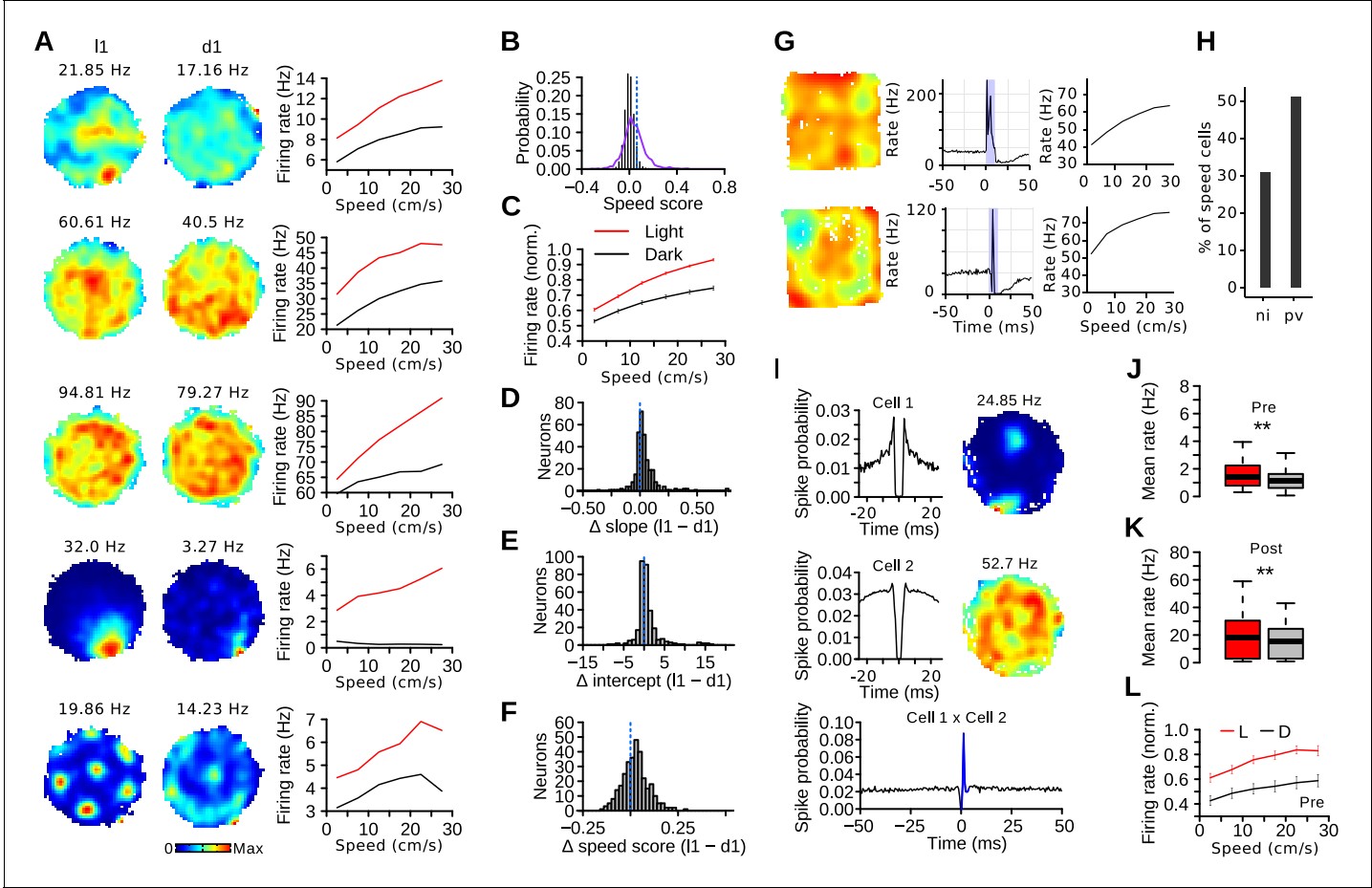

**Figure 4.** Visual stimuli alter the MEC speed code. (A) Examples of firing rate maps and speed tuning curves during light and dark trials for cells with a significant speed score. (B) Real (purple line) and surrogate (solid black bars) distributions of speed scores from MEC neurons. The dotted blue line indicates the threshold for statistical significance. (C) Mean normalized firing rate (± s.e.m) of speed-modulated cells as a function of running speed during l1 and d1 trials. (D, E and F) Difference of speed-rate slopes, intercepts and speed scores of speed-modulated cells during l1 and d1 trials. The dotted lines indicates chance levels. (G) Firing rate map, response to laser stimulation and speed tuning curve of two PV-expressing neurons. (H) Percentage of speed-modulated cells in PV-expressing neurons (pv) and in non-identified MEC neurons (ni). (I) Example of a putative excitatory connection involving a post-synaptic speed cell. Left: spike-time autocorrelations of putative pre- (top) and post-synaptic (middle) neurons. Right: firing rate maps during light trials. Bottom: spike-time crosscorrelation of the two neurons. The blue color indicates the peak detection period. (J) Mean firing rate during light and dark trials of putative presynaptic neurons with excitatory interactions with a speed cell. (K) Mean firing rate during light and dark trials of speed cells receiving putative excitatory connections from a local neuron. (L) Mean firing rate (± s.e.m) as function of running speed of putative presynaptic neurons with excitatory interactions with a speed cell. **p<0.01.

The presence of visual cues changed the speed code (*Figure 4A*). Most speed-modulated cells had a lower firing rate at a given speed during dark trials (*Figure 4C*, at 20–25 cm/s, 222 out of 304 cells, contingency chi-square test, $\lambda^2 = 64.47$, *df* = 1, p<$10^{-16}$). The slopes of the regression line between speed and firing rate were steeper during light than dark trials (*Figure 4D*; paired Wilcoxon signed rank test, *n* = 304, median, light: 0.065, dark: 0.038, v = 35346, p<$10^{-15}$). The intercepts (predicted firing rate during immobility) were also higher during light than dark trials (*Figure 4E*; median, light: 2.621, dark: 2.428, v = 31050, p<$10^{-7}$). Speed scores were higher during light than dark trials (*Figure 4F*; median, light: 0.086, dark: 0.060, v = 32735, p<$10^{-10}$). These changes in slopes, intercepts and speed scores were also present when considering only neurons recorded from MEC tetrodes (*n* = 144 speed-modulated cells, slope: p<$10^{-13}$, intercept: p=0.0017, speed score: p<$10^{-9}$) or when using the median score of each mouse as a statistical unit (*n* = 8 mice, slope: p=0.0078, intercept: p=0.016, speed score: p=0.039). The rate modulation by visual stimuli was sufficiently large that speed-modulated cells had a lower mean firing rate during dark trials (paired Wilcoxon signed rank test, *n* = 304, median, light: 3.47 Hz, dark: 2.89 Hz, v = 33797, p<$10^{-12}$) even though running speed was higher during these trials (paired Wilcoxon signed rank test, *n* = 89 recording sessions, median, light: 12.70 cm/s, dark: 14.81 cm/s, v = 124, p<$10^{-14}$). The median change of firing rate from d1 to l1 trials, expressed as a percentage, was 12.36%. Thus, the rate response of speed-modulated neurons is determined by both internal self-motion cues and visual information.

Several speed-modulated cells have a high firing rate, which suggests that these might be PV-expressing MEC interneurons (*Buetfering et al., 2014*; *Kropff et al., 2015*). We tested whether PV-expressing interneurons were more likely to be classified as speed cells than other MEC neurons by re-examining the data from Buetfering and co-workers (*Buetfering et al., 2014*). The activity of 140 optogenetically-identified PV-expressing interneurons was examined alongside that of 996 other MEC neurons (*Figure 4G*). The speed score was used to identify speed-modulated cells (380 out of 1136 neurons, threshold = 0.079) in mice exploring an open field. The proportion of speed modulated cells was higher in the PV population than in other non-identified MEC neurons (*Figure 4H*; PV-expressing: 72 speed cells out of 140 neurons, 51.4%, other neurons: 308 out of 996 neurons, 27.1%, contingency chi-square test, $\lambda^2 = 22.27$, *df* = 1, p<$10^{-6}$).

PV-expressing MEC neurons receive strong excitatory inputs from local neurons (*Couey et al., 2013*; *Pastoll et al., 2013*; *Buetfering et al., 2014*). We therefore investigated if the reduction in the firing rate of some high firing rate speed-modulated cells in darkness could be explained by changes in local excitatory inputs. Putative monosynaptic excitatory connections between MEC neurons were identified from spike-train crosscorrelations (*Figure 4I*) (*Royer et al., 2012*; *Buetfering et al., 2014*). Out of 10,880 crosscorrelograms, 61 (0.56%) showed a low latency peak indicative of excitatory connections. All connected pairs were made up of neurons recorded on the same tetrode. The percentage of putative excitatory connections for pairs of cells recorded on the same tetrode was 1.85 %.

Within functionally coupled neurons, speed-modulated cells were more likely to be post-synaptic (24 out of 42) than pre-synaptic (18 out of 58 neurons) neurons ($\lambda^2 = 5.7868$, *df* = 1, p=0.0162). The firing rate of the cells pre-synaptic to speed-modulated cells was lower during dark than light trials (*Figure 4J*; paired Wilcoxon signed rank test, *n* = 37, v = 555, p=0.0016). The firing rate of the post-synaptic speed-modulated cells was also lower during dark trials (*Figure 4K*; *n* = 24, median, light: 18.44 Hz, dark: 15.44 Hz, v = 264, p=0.00057). Moreover, the firing rate of pre-synaptic neurons increased with running speed (*Figure 4L*; paired Wilcoxon signed rank test, difference rate 2.5 cm/s vs 27.5 cm/s, *n* = 37 light: v = 113, p=0.00016, dark: v = 109, p=0.00012). Thus, the change in firing rate of some speed-modulated interneurons in darkness can be explained by a reduced local excitatory input.

## Impaired border representation in darkness

Border cells are thought to anchor grid cell fields to the geometry of an environment (*Barry et al., 2007*; *Solstad et al., 2008*; *Lever et al., 2009*; *Evans et al., 2015*; *Hardcastle et al., 2015*). We assessed the effect of visual landmarks on the firing of border cells. The activity of border cells during light trials was restricted to the periphery of the circular arena and usually covered less than half of the total circumference. In darkness, the activity of several border cells was no longer limited to the periphery of the arena (*Figure 5A*). We quantified how restricted the firing of border cells was

to the periphery of the arena (DM, see Experimental Procedures). The firing of border cells was less concentrated at the periphery during d1 trials (*Figure 5B*; paired Wilcoxon signed rank test, $n = 63$ border cells, $v = 271$, $p<10^{-7}$). In addition, the polarity of the firing maps was reduced during d1 trials compared to l1 trials (see Materials and methods, *Figure 5C*; $v = 1698$, $p<10^{-6}$). Similar findings were observed when limiting the analysis to border cells recorded from MEC tetrodes ($n = 38$ border cells, DM: $v = 156$, $p=0.0014$, map polarity: $v = 552$, $p=0.008$). When using the median of each mouse as a statistical unit, we observed a significant reduction of DM in darkness ($n = 7$ mice, $v = 1$, $p=0.031$), but the reduction in map polarity in darkness did not reach significance level ($n = 7$ mice, $v = 24$, $p=0.11$).

## Reduced head-direction selectivity in darkness

A total of 85 head-direction cells were recorded on the circular arena. We quantified changes in head-direction selectivity using the mean vector length of the rate/head-direction histograms of head-direction cells. As shown in *Figure 6A*, most head-direction cells had reduced head-direction selectivity during dark trials compared to light trials. Head-direction vector length of head-direction cells were lower during d1 trials than l1 trials (*Figure 6B*, paired Wilcoxon signed rank test, $v = 3617$, $p<10^{-15}$). A similar conclusion was reached when limiting the analysis to MEC tetrodes ($n = 54$ head-direction cells, $v = 1457$, $p<10^{-10}$) or when using the median of each mouse as a statistical unit ($n = 8$ mice, $v = 36$, $p=0.0078$).

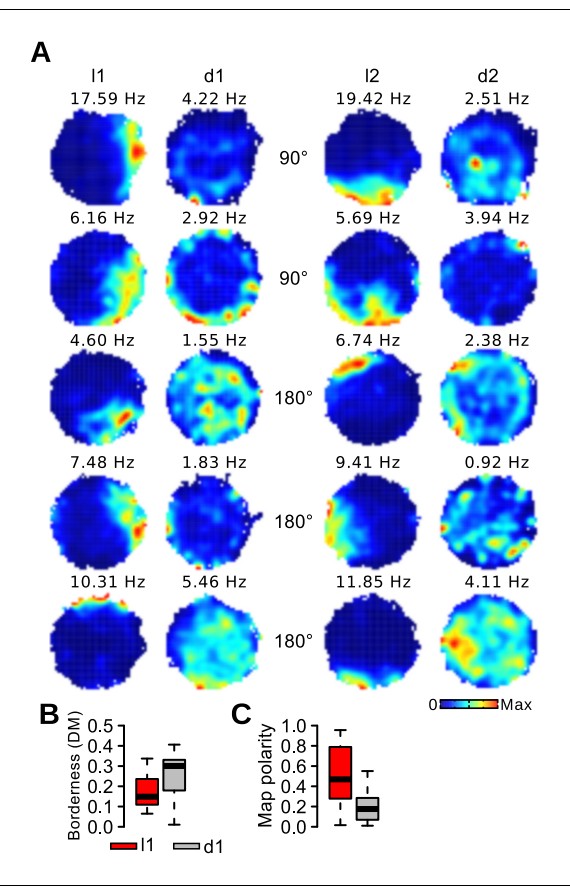

**Figure 5.** Impairment of border representation in darkness. (**A**) Firing maps of 5 border cells during light and dark trials. (**B**) Borderness (DM) of the firing rate maps of border cells during light and dark trials. (**C**) Polarity of the firing rate maps of border cells during light and dark trials.

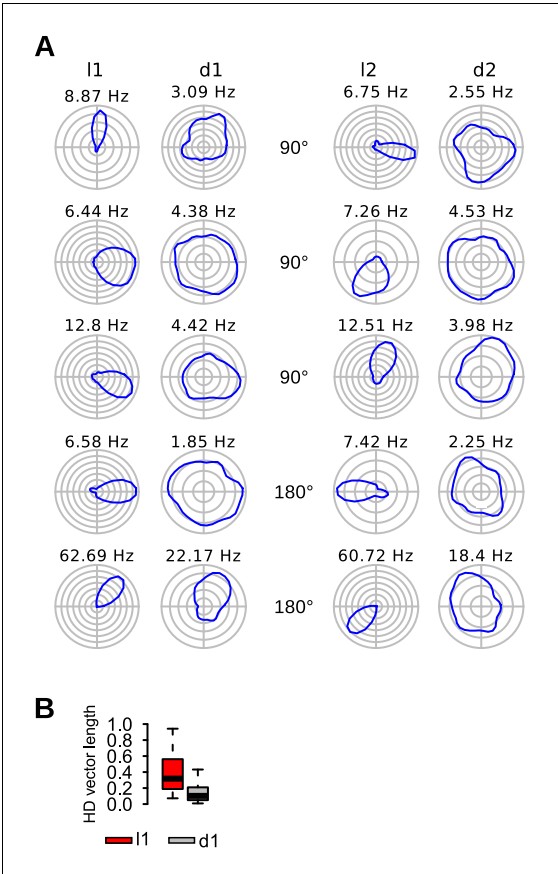

**Figure 6.** Reduced head-direction selectivity in darkness. (**A**) Firing maps of 5 head-direction cells during light and dark trials. (**B**) Head-direction vector length of head-direction cells during l1 and d1 trials.

## Firing rate changes in MEC neurons between light and dark trials

We next investigated whether the firing rate of MEC neurons changed significantly when visual landmarks were eliminated. For each neuron, a rate discrimination index was obtained using: $(\lambda_{light} - \lambda_{dark})/(\lambda_{light} + \lambda_{dark})$, where $\lambda_{light}$ and $\lambda_{dark}$ are the mean firing rate during light and dark trials, respectively. Significance levels were obtained on a cell-by-cell basis by shuffling trial identities 500 times to obtain a distribution of discrimination indices. The 99th percentiles of the surrogate distributions served as significance levels. Only periods during which the running speed of the mice was between 5 and 20 cm/s were used. Examples of neurons with a significant rate change between light and dark trials are shown in *Figure 7A*. For some neurons, the firing rate changes between trial types were readily visible in their instantaneous firing rate (*Figure 7B*). When all recorded neurons were considered, the median rate discrimination index (0.072) was significantly larger than 0, demonstrating that MEC neurons tend to have higher firing rates when visual landmarks are present (*Figure 7C*, paired Wilcoxon signed rank test, $n = 880$, $v = 289120$, $p < 10^{-16}$). This rate change was also significant when only neurons recorded from MEC tetrodes were considered ($n = 447$, $v = 76177$, $p < 10^{-16}$) or when the median rate discrimination index of each mouse was used as a statistical unit ($n = 8$ mice, $v = 36$, $p = 0.0078$). Out of 880 MEC neurons, 503 (57.2%) significantly changed their firing rate between light and dark trials. These neurons were more likely to reduce (76.74%) than increase (23.26%) their firing rate in darkness ($\chi^2 = 143.86$, $df = 1$, $p < 10^{-16}$). The proportion of significant neurons in the different functional cell types is shown in *Figure 7D*. Border cells were more likely than grid cells to change their firing rate depending on the presence of visual landmarks ($\chi^2 = 11.845$, $df = 5$, $p = 0.037$).

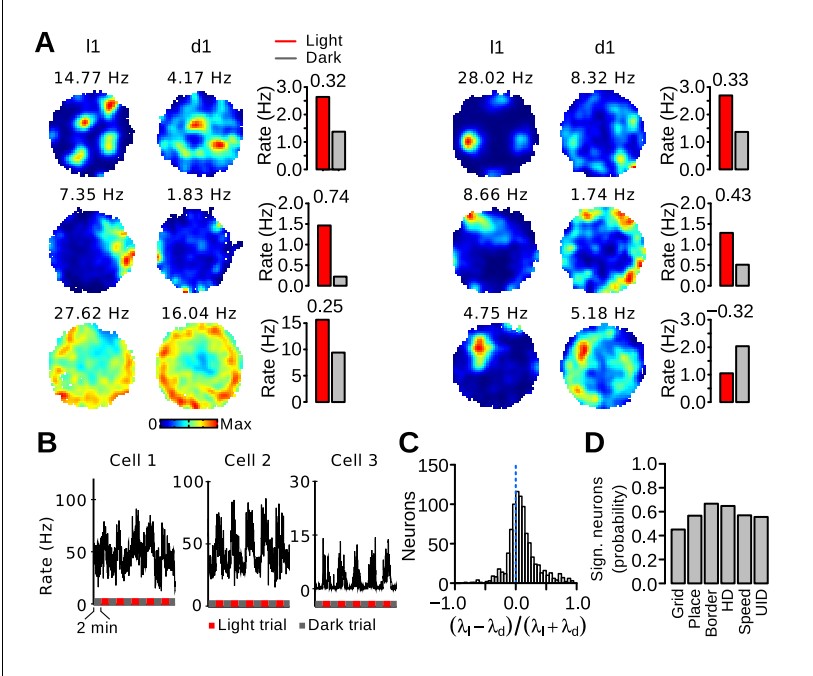

**Figure 7.** Firing rate changes of MEC neurons between light and dark trials. (A) Firing maps and mean firing rates during light and dark trials for 6 neurons. The number above each bar plot is the rate discrimination index. (B) Examples of instantaneous firing rates of 3 neurons during light and dark trials (smoothing kernel width s.d = 900 ms). (C) Distribution of the rate discrimination indices for all recorded neurons (including putative interneurons). Most neurons had a positive discrimination index, indicating higher firing rates during light trials. The dotted line indicates chance level. (D) Proportion of neurons with a significant rate change between light and dark trials in different functionally defined cell types (Grid: grid cells, Place: irregular spatially selective cells, Border: border cells, HD: head-direction cells, Speed: speed-modulated cells, UID: unidentified cells).

We also tested whether putative interneurons (cells with a mean firing rate > 10 Hz, $n$ = 133) and putative principal cells (cells with a mean firing rate < 5 Hz, $n$ = 694) were equally likely to change their mean firing rate depending on the presence of visual landmarks. The probability of observing significant rate change between light and dark trials was similar for interneurons and principal cells (interneurons: 0.617, principal cells: 0.549, $\lambda^2$ = 1.802, $df$ = 1, p=0.18).

## Firing rate changes associated with nonmetric contextual cue manipulation in 1D environment

The results presented so far indicate that the firing rate of MEC neurons is modulated by the presence of visual landmarks. This suggests that the firing rate of spatially selective neurons in the MEC could also convey information about the distinct visual stimuli present in an environment. We therefore performed a second experiment with a new cohort of mice to test whether the rate code of MEC neurons discriminates different nonmetric visual landmarks (or contexts) in an otherwise unchanged environment. The activity of 479 neurons (5 mice, 48 recording sessions) was recorded as mice ran back and forth on a linear track flanked by two side walls (*Figure 8A*). Histological analysis indicated that 14 recording sites were in the MEC, 4 in the MEC/parasubiculum border, and 13 in the parasubiculum (*Figure 8—source data 1*, *Figure 8—source data 2*). There were 3 lighting conditions (l1, l2 and d). In l1, a single row of 48 LEDs on one wall was turned on. In l2, 4 rows of 12 LEDs located on the opposite wall were turned on. In d, all LEDs were turned off. The condition was changed randomly every 5th run. Thus, the only difference between l1 and l2 conditions was the LED pattern that was turned on and the geometry of the apparatus remained unchanged.

A 20-min exploration trial in a square open field preceded the linear track trials and served to identify the different functional cell types (*Figure 8B*). We recorded 138 grid cells, 28 border cells

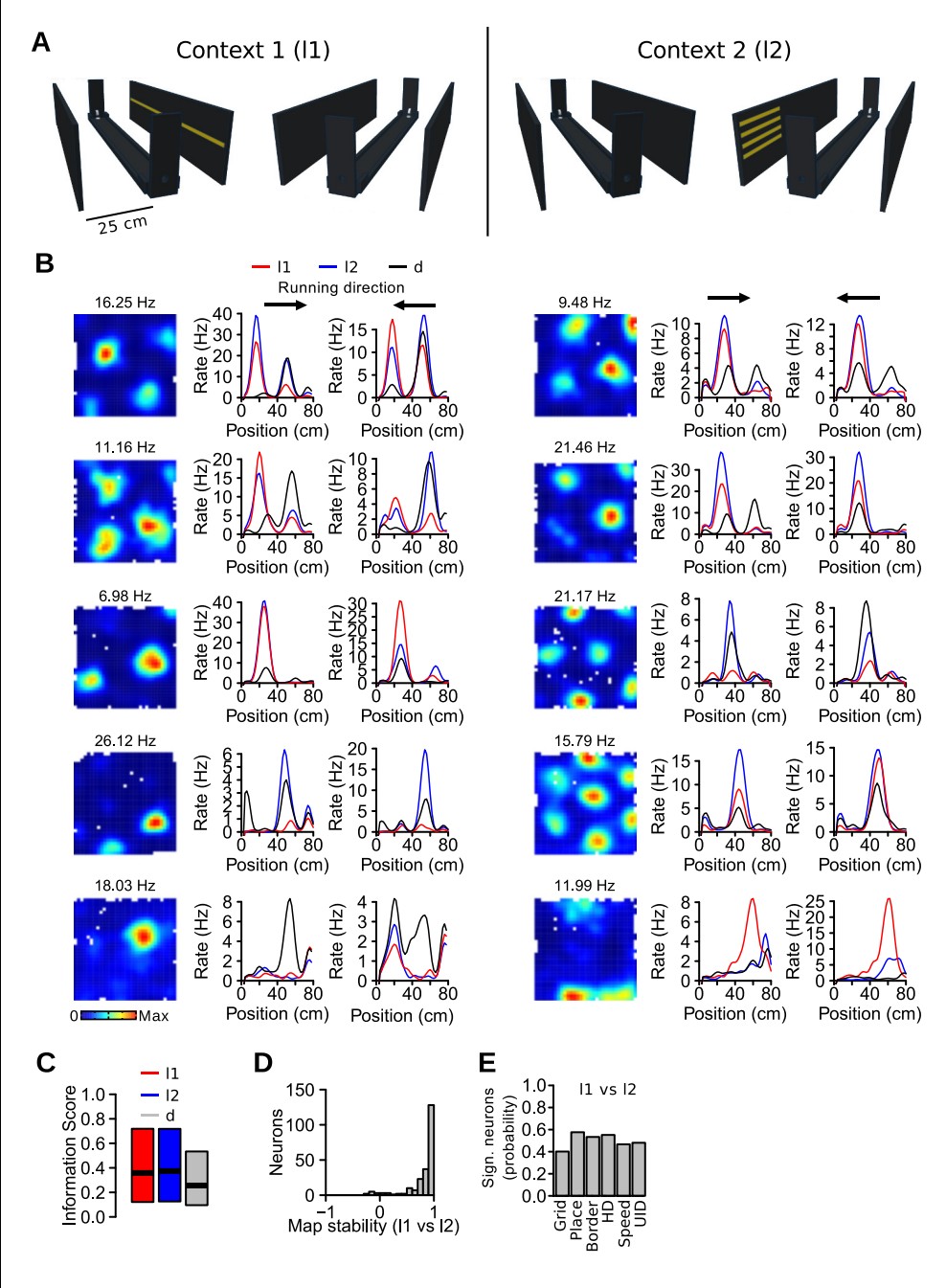

**Figure 8.** Nonmetric contextual visual cues affect firing rates of MEC neurons in a 1D environment. (**A**) Schematic of the linear track and side walls. The linear track was flanked by two side walls on which LED arrays were attached. In the first context (l1, left), a single row of LEDs on one wall was turned on. In the other context (l2, right), 4 shorter rows of LEDs on the opposite wall were turned on. All LEDs were turned off during dark (d) trials. (**B**) Example of neurons with firing rate changes between the different conditions. First column, firing rate maps in the square open field. Next two columns, linear firing rate maps for the 3 conditions, plotted separately for each running direction. Note that the range of the y-axes varies. (**C**) Spatial information scores (median, first and third quartiles) in the 3 conditions. (**D**) Stability of linear firing rate maps (correlation coefficient) between l1 and l2 conditions. (**E**) Proportion of neurons with a significant rate change between l1 and l2 across functionally defined cell types (Grid: grid cells, Place: irregular spatially selective cells, Border: border cells, HD: head-direction cells, Speed: speed-modulated cells, UID: unidentified cells).

*Figure 8 continued on next page*

*Figure 8 continued*

The following source data is available for figure 8:

**Source data 1.** Histological results of the mice recorded on the linear track.
**Source data 2.** Location of the tetrode tips in each hemisphere for recordings done on the linear track.

and 133 irregular spatially selective cells on the track. Linear firing rate maps were calculated for the 3 lighting conditions, plotting the maps with different running direction separately. *Figure 8B* shows examples of spatially selective neurons with clear firing rate changes between the 3 conditions (l1, l2 and d).

A quantitative analysis of the linear firing rate maps was performed, including only grid cells, border cells and irregular spatially selective cells. Spatial information scores were not significantly different between l1 and l2 conditions (*Figure 8C*, paired Wilcoxon signed rank test, $n$ = 299 neurons, $v$ = 21632, p=0.60), but were lower in darkness (l1 vs d, $v$ = 34469, p<$10^{-16}$). For most neurons, the location of the firing fields appeared to be preserved across the 3 conditions. Indeed, the median correlation coefficient between linear rate maps of l1 and l2 was 0.92 (*Figure 8D*). The correlation coefficients were slightly lower between l1 and d (median: 0.81, paired Wilcoxon signed rank test, l1-l2 vs l1-d r values: $v$ = 31909, p=$10^{-10}$). Thus, there was no major reorganization of the firing fields between conditions.

To identify neurons with significant firing rate changes between conditions, a shuffling procedure in which the identity of the conditions was reassigned randomly 500 times was used to obtain chance levels for rate differences. The difference observed at each bin of the linear rate maps was compared to those of the surrogate distribution. Neurons with more than 5 bins in which the observed difference had a probability below 0.01 were considered significant. Approximately half of all recorded neurons showed significant changes between l1 and l2 conditions (245 out of 479, 51.1%). The proportion of neurons with significant rate changes was the same across all functionally defined cell types (*Figure 8E*, $\lambda^2$ = 9.259, $df$ = 5, p=0.10). When considering only neurons recorded from MEC tetrodes, 35.3% (24 out of 68) of the neurons showed significant differences between l1 and l2 conditions. The percentage of neurons with significant rate changes between l1 and d conditions was 70.8% (339 out of 479) when considering all recorded neurons and 50% (34 out of 68) when limiting the analysis to neurons recorded from MEC tetrodes. These results demonstrate that the firing rate of MEC neurons along the linear track conveys information about distinct nonmetric contextual cues.

## Discussion

Our study aimed at establishing the contribution of visual landmarks to the firing of spatially selective neurons in the MEC. The principal results indicate that visual information 1) prevents a rapid destabilization of the grid cell firing pattern, 2) changes the speed code of MEC neurons, and 3) modulates the firing rate of neurons of all principal MEC cell types.

We found that during dark trials the periodic pattern of grid cells rapidly became unstable. This instability was greater than previously reported (*Hafting et al., 2005*; *Allen et al., 2014*). Indeed, our analysis indicates that instability in the grid pattern was detectable within the first few seconds in darkness. Several factors could contribute to this grid instability. First, the circular arena had no wall in the current experiment. This eliminates a potential source of uncontrolled olfactory or tactile cues that the animal can use to anchor its spatial representation in darkness. Second, two polarizing LED panels were used during the recording sessions and repeatedly caused reorientations of the MEC spatial representations. The spatial relationships between the grid pattern and uncontrolled external cues were therefore not stable over time. This might have reduced the contribution of uncontrolled external cues to the anchoring of MEC spatial representations during dark trials. Third, it has also been suggested that border cells reset cumulative error in the grid cell network when an animal encounters a boundary (*Hardcastle et al., 2015*). Thus, the altered spatial representation by border cells in darkness might have contributed to grid cell instability. Our findings indicate that in environments deprived of salient landmarks, error in the estimation of position by grid cells accumulates faster than previously thought.

A population of MEC neurons encodes the running speed of the animal (*Kropff et al., 2015*). It is thought that this speed code in the MEC remains unchanged across different contexts (*Kropff et al., 2015*). Our findings, instead, point to an important contribution of visual cues, and perhaps optic flow, to the speed code, as most speed-modulated neurons had a lower firing rate in darkness. Speed-modulated neurons included high firing rate interneurons but also a significant proportion of grid cells and other spatially selective neurons. GABAergic neurons expressing parvalbumin were more likely to have high speed scores compared to other MEC neurons. The analysis of functional connections onto these GABAergic neurons indicated that cells pre-synaptic to speed cells decreased their firing rate during dark trials, providing a simple explanation for the reduction in firing rate of GABAergic speed cells. As speed-modulated neurons have been observed in several brain regions (*McNaughton et al., 1983*; *Saleem et al., 2013*; *Bender et al., 2015*; *Fuhrmann et al., 2015*), it will be important to establish which pathways are required for the speed modulation of MEC neurons and whether these afferent speed signals are also modulated by visual landmarks.

We found that a significant proportion of MEC neurons changed their firing rate when visual landmarks were eliminated. This demonstrates that the level of activity in the MEC network during navigation is not constant but varies depending on the visual cues present in an environment. Although cells of all functionally defined cell types responded to light, border cells appear especially sensitive to manipulation of visual inputs. An important cortical afferent carrying visual information to the MEC originates in the postrhinal cortex (*Burwell and Amaral, 1998a*, *1998b*) and has a depolarizing effect on principal neurons of the MEC (*Koganezawa et al., 2015*). It is tempting to speculate that this projection is more active when visual landmarks are present. This could explain why the firing rate of MEC neurons was generally lower in darkness.

We found that approximately half of the recorded grid cells changed their firing rate during 1D navigation depending on nonmetric contextual cues. This demonstrates that the rate code of grid cells in 1D environments is context specific. It has recently been proposed that grid cells in rats trained to perform a conditional discrimination task can encode object information (*Keene et al., 2016*). Our findings complement this study by showing that changes in firing rate caused by contextual information do not strictly depend on the behavioral demands of the task. Indeed, on the linear track the mice did not have to change their behavior following contextual changes. Inclusion of contextual information in the rate code of grid cells might take place continuously during 1D navigation. The rate change of MEC neurons associated with distinct contexts on the linear track is in agreement with $Ca^{2+}$ activity changes observed in MEC Reelin$^+$ cells following exploration of different enclosures (*Kitamura et al., 2015*). The results of the current study reveal the functional cell types of the context-specific neurons.

The mechanisms behind the rate changes of grid cells during 1D navigation or memory tasks are still not clear. One hypothesis is that the location of the grid fields remains stable but that visual inputs onto grid cells alter their in-field firing rates. Such rate changes within stable fields have already been observed in hippocampal place cells during 2D navigation (*Leutgeb et al., 2005*) or during different memory tasks (*Lipton et al., 2007*; *Ferbinteanu et al., 2011*; *Allen et al., 2012*; *Ainge et al., 2012*). An alternative hypothesis is that the rate changes reflect a shift in the location of the grid firing fields caused by the nonmetric contextual cues (*Marozzi et al., 2015*). If one considers 1D firing maps on the track as slices through the 2D map of the neurons (*Yoon et al., 2016*), the 1D maps of grid cells in two contexts on the linear track could be considered as slices with different phases or orientations. It is also possible that the contextual cues affect the periodicity of the underlying 2D grid lattice (*Krupic et al., 2015*; *Stensola et al., 2015*). Both hypotheses remain plausible at this time, but the high correlations between the linear firing rate maps in the two contexts indicate that the change of field location was minimal. Regardless of the precise mechanisms, the outcome is that the rate code of grid cells along 1D paths is context dependent. Since navigation in the real world is often confined to preexisting paths, the firing rate of grid cells in these conditions provides information not only about the position of the animal but also about the visual cues encountered along the path.

# Materials and methods

## Surgical procedure

All experiments were carried out in 3–6 month-old male wild type C57BL/6 mice and were approved by the Governmental Supervisory Panel on Animal Experiments of Baden-Württemberg in Karlsruhe (35–9185.81/G-50/14). Mice were singly housed and kept on a 12 hr light-dark schedule with all procedures performed during the light phase. The mouse cages were 26 cm long, 20 cm wide and 14 cm high. The cage floor was covered with 2 cm of saw dust and 1–2 facial tissues were placed in the cage. Mice were implanted with 4 movable tetrodes in each hemisphere. The tetrodes were constructed from 4 12 µm-diameter tungsten wires (California Fine Wire Company, Grover Beach, California) and held in a microdrive assembly that allowed them to be moved individually. Before implantation, tetrodes were gold plated to reduce their impedance to 300–500 kΩ.

Mice were anesthetized with isoflurane (1–3%) and placed into a stereotaxic apparatus. The skull was exposed and four anchor screws were inserted into the skull. Two screws located above the cerebellum served as ground and reference signals. The following coordinates were used (ML: ± 3.1 mm from bregma, AP: 0.2 mm from transverse sinus, 6° in the posterior direction). The tetrodes were lowered into the cortex and the microdrive was fixed to the skull with dental cement. Mice were given one to two weeks to recover after surgery.

## Recording system, spike extraction and spike clustering

The animals were connected to the data acquisition system (RHD2000-Series Amplifier Evaluation System, Intan Technologies, analog bandwidth 0.09–7603.77 Hz) via a lightweight cable and the signal was sampled at 20 kHz. Action potentials were detected off-line from the bandpass-filtered signal (800–5000 Hz). Waveform parameters were obtained from a principal component analysis and clusters of spikes were automatically generated using Klustakwik. Spike clusters were refined manually with a graphical interface program. Cluster separation quality was assessed from the spike-time autocorrelation and isolation distance. A refractory period ratio was calculated from the spike-time autocorrelation (from 0 to 25 ms, bin size: 0.5 ms). The mean number of spikes from 0 to 1.5 ms was divided by the maximum number of spikes in any bin between 5 and 25 ms. Clusters with a refractory period ratio larger than 0.125 were not kept. In addition, clusters with an isolation distance (*Schmitzer-Torbert et al., 2005*) shorter than 5 were excluded from the analysis.

Two infrared-LEDs (wave length 940 nm), one large and one small, were attached to the headstage. The large and small LEDs were located ahead and behind the head of the animal, respectively, with a distance of 8 cm between their centers. An infrared video camera (resolution of 10 pixels/cm, DMK 23FM021, The Imaging Source) monitored the LEDs at 50 Hz. The location and head direction of the animal were tracked on-line with custom software.

## Initial training

After the recovery period, mice were put on a food restriction diet to reduce their weight to 85% of their normal free-feeding weight. They were then trained 3 times a day (3 × 10 min) to run in a 70 × 70 cm open field to retrieve food rewards (AIN-76A Rodent tablets 5 mg, TestDiet) delivered at random locations from pellet dispensers located above the ceiling of the recording environment (CT-ENV-203-5 pellet dispenser, MedAssociates). The pellet dispensers were controlled by a microcontroller (Arduino Uno) and the inter-delivery intervals ranged from 20 to 40 s.

After 2 days of training, the procedure continued (3 × 15 min) but the mice were connected to the recording system. The tetrodes were lowered on each day and the raw signals were monitored on an oscilloscope. Recordings began when large theta oscillations were observed on most tetrodes (*Fyhn et al., 2008*). The tetrodes were also lowered by approximately 25–50 µm at the end of each recording session.

## Circular arena

The apparatus consisted of an elevated (4.5 cm) gray circular PVC platform (80 cm diameter) located in the center of a gray square box (100 × 100 × 19.5 cm). The square box was filled with water up to 3 cm to prevent the animal from getting off the circular arena. The recording environment was surrounded by opaque black curtains. On every side of the box, a LED panel provided a polarizing

cue to the animal (90° angle between the lights). The light panels consisted of black aluminum sheets (46 cm × 33 cm) with two horizontal LED strips (45 cm long, 25 cm apart from each other; color temperature: 3000 K, Ribbon Slim Top, Ledxon Group, powered by 6 1.2 V batteries). These light panels were the only potential source of visible light in the recording environment. An audio speaker located directly above the arena emitted a white noise, overshadowing uncontrolled auditory cues.

After initial training, the mouse was transported from the holding room to the recording room in an opaque circular container (15 cm diameter) and underwent a disorientation procedure in which the container was spun 5 times in clockwise and 5 times in counterclockwise direction (approximately 1 rotation/s). The mouse was connected to the recording system outside of the curtains, and it was carried within the holding box into the enclosure and placed on the arena. After closing the curtains around the apparatus, the recording started and the experimenter left the room for the duration of the recording session. The mouse had no prior experience of the recording room before the first recording session.

The light panels were controlled by a microcontroller (Arduino Uno) via a 4 channel relay module. For each recording session, two of the four light panels were chosen randomly and only these two lights were used during the recording session (referred to as l1 and l2). The location of l1 and l2 varied randomly across days in the same animal. A session started with a baseline of 10 min with l1 turned on, followed by 2 min in darkness and a series of 60 2-min trials, alternating between light and dark trials. Only one light was switched on during a given light trial, and the order of presentation of the two lights was random. At the end of the light-dark trial sequence, an additional baseline of 10 min with l2 turned on was performed. The experimenter entered the recording room, the mouse was removed from the recording environment and the tetrodes were lowered by approximately 25–50 μm. To ensure that mice visited every location on the elevated arena, food rewards were delivered at random positions during recording sessions.

## Identification of spatially selective neurons on the circular arena

Data analysis was performed in the R software environment and the source code is available at https://github.com/kevin-allen/prog_perez_escobar_2016. Spatial firing rate maps were generated by dividing the environment into 2 × 2 cm bins. The time spent in each bin was calculated and the resulting occupancy map was smoothed with a Gaussian kernel (s.d = 3 cm). The number of spikes emitted as the animal was in each bin was divided by the corresponding bin of the occupancy map to obtain the firing rate map, which was then smoothed with a gaussian kernel function (s.d = 3 cm). Only periods when the mouse ran faster than 3 cm/s were considered. The spatial information score (*Skaggs et al., 1996*) was defined as follows:

$$I = \sum_{i=1}^{N} p_i \frac{\lambda_i}{\lambda} log_2 \frac{\lambda_i}{\lambda}.$$

where $p_i$ is the occupancy probability of bin $i$ in the firing map, $\lambda_i$ is the firing rate of bin $i$, and $\lambda$ is the mean firing rate of the neuron.

Spatial autocorrelations were calculated from the firing rate maps. Peaks in the autocorrelation matrix were defined as > 10 adjacent bins with values > 0.1. The 60° periodicity in the spatial autocorrelation matrix was estimated as follows (*Sargolini, 2006*). A circular region of the spatial autocorrelation matrix containing up to six peaks and excluding the central peak was defined. Pearson correlation coefficients (*r*) were calculated between this circular region of the matrix and a rotated version of itself (by 30°, 60°, 90°, 120°, and 150°). A grid score was obtained from the formula:

$$\left( \frac{r_{60°} + r_{120°}}{2} \right) - \left( \frac{r_{30°} + r_{90°} + r_{150°}}{3} \right).$$

Significance thresholds for information and grid scores were obtained by shifting the position data by at least 20 s before recalculating both scores. This procedure was repeated 100 times for each neuron in order to obtain surrogate distributions. The 95th percentiles of the null distributions were used as significance thresholds. Neurons with a significant grid score during both 10-min baselines or during one baseline and l2 trials were defined as grid cells. The spacing of a grid cell was defined as the mean distance from the central peak to the vertices of the inner hexagon in the spatial autocorrelation.

Border cells were identified using a border score calculated from two variables ($CM_{0.5}$ and DM). The pixels of a firing rate map that were directly adjacent to the periphery of the arena were identified. Firing fields, defined as groups of adjacent pixels with a firing rate larger than 20% of the peak firing rate of the map and covering at least 40 cm$^2$, were detected. For each field, the proportion of the pixels along the periphery that were also part of the field was calculated. CM was defined as the maximum proportion obtained over all possible fields. Because the firing fields of a border cell in circular environment typically cover up to half of the periphery, the variable $CM_{0.5}$ was defined as $(1 - |(0.5 - CM)| * 2)$. $CM_{0.5}$ had a value of 1 when the firing field covered half of the periphery and a value of 0 when the field covered all of the periphery or nothing of the periphery. DM was the mean shortest distance to the periphery for pixels that were part of a firing field, weighted by the firing rate in each pixel. DM was then normalized as follows. For each pixel in the map, the shortest distance to the periphery was calculated. The largest value obtained over all map pixels was the value used for the normalization. The border score was defined as $(CM_{0.5} - DM)/(CM_{0.5} + DM)$. Significance level was obtained with the same shuffling procedure as for information and grid scores. Border cells were defined as cells with a significant border score during both baselines or during one baseline and l2 trials.

As most border cells in a circular environment cover only a section of the periphery (*Solstad et al., 2008*), we also calculated a polarity score for each map. For each pixel of the map, a vector was created. Its direction was that of the pixel relative to the center of the map and its length was set to the firing rate in that pixel. The map polarity score was defined as the length of the resulting vector after summing individual vectors and normalizing the length by the sum of the firing rate of all pixels in the map.

Irregular spatially selective cells were neurons that were not classified as grid or border cells and that had a significant spatial information score during both baselines or during one baseline and l2 trials.

Head-direction cells were identified by constructing a histogram with the firing rate of a neuron as a function of head direction (10° per bin). The mean vector length of the histogram was used as a measure of head-direction selectivity. A null distribution of mean vector length was obtained by shifting the head-direction data by at least 20 s before recalculating the mean vector length. This shuffling procedure was repeated 100 times for each neuron and the 95th percentile of the surrogate distribution served as significance level. Head direction cells had a significant vector length and a peak firing rate above 5 Hz during both baselines or during one baseline and l2 trials.

To identify speed-modulated cells, the instantaneous firing rate of the neurons was calculated. The number of spikes in 1 ms time windows was counted and a convolution between this spike count array and a Gaussian kernel (s.d = 100 ms) was performed. The resulting vector was integrated over 100 ms time windows. Periods during which the mice ran slower than 3 cm/s were removed from the analysis. Running speed was estimated every 20 ms. A speed score was defined as the Pearson correlation coefficient between the instantaneous firing rate and the running speed of the animal. Chance levels were obtained with a shuffling procedure ($n = 100$) in which the speed vector was shifted by at least 20 s before calculating speed scores. To be considered a speed-modulated cell, the neuron had to have a speed score above the 95th percentile of the surrogate distribution during both baselines or during one baseline and l2 trials.

## Stability of firing rate maps

To assess the stability of the spatial firing patterns during dark trials, we divided d1 trials into 12 blocks of 10 s. Maps were constructed for the 12 blocks, concatenating homologous blocks across trials of a given condition. The stability of the map in each block was obtained by calculating the correlation coefficient between the block-specific firing map and the map containing all 120-s l1 trials. As controls, the l1 trials were also divided into 12 blocks and the maps observed during each block were compared to the maps containing all 120-s l1 trials.

## Spike distance metric

Error accumulation in the spikes of grid cells was estimated by the spike distance metric (SDM) (*Hardcastle et al., 2015*). For each cell, a firing rate map was constructed using the first 60 s of l1 trials. A firing field was defined as an area of at least 20 cm$^2$ in which each bin had a firing rate above

the 75th percentile of the rate distribution of all bins in the map. The center of mass of each firing field was calculated. The radius of a firing field was equal to the radius of a circle with an area equal to that of the firing field. For each spike, the distance of the animal location at the time of the spike to the closest firing field center of mass was calculated. SDM was this distance divided by the mean radius of all detected fields in the firing rate map. SDM was calculated for spikes fired in the last 60 s of l1 trials and the entire d1 trials.

## Distance coding by grid cells

Spike-triggered firing rate maps were constructed by taking each spike of a neuron as a reference spike and considering only data of the next 10 s. The position data within each time window were shifted so that the reference spike was aligned to position (0,0). The space surrounding the reference spikes was divided into $2 \times 2$ cm bins and both the occupancy maps and the resulting spike-triggered firing rate maps were smoothed with a Gaussian kernel (s.d = 2 cm). To obtain a distance tuning curve, the bins of the spike-triggered firing rate map were used to calculate the mean firing rate as a function of distance from the reference spikes. The distance was normalized by dividing it by the spacing of the grid cell measured during the first baseline of the recording session. A distance score was defined as $(\lambda_1 - \lambda_{0.5})/(\lambda_1 + \lambda_{0.5})$, where $\lambda$ indicates the firing rate of a neuron at a given normalized distance. Chance levels for distance scores were obtained by shifting the spike trains by at least 20 s relative to the position data before recalculating the scores. Only grid cells with a spacing shorter than 50 cm were used in the distance analysis.

## Detection of putative excitatory connections based on spike-time crosscorrelations

Putative excitatory connections were detected as narrow peaks in the spike-time crosscorrelations of simultaneously recorded neurons (from −50 to 50 ms, bin size = 0.5 ms) (*Csicsvari et al., 1998*; *Marshall et al., 2002*; *Maurer et al., 2006*; *Mizuseki et al., 2009*; *Latuske et al., 2015*). The crosscorrelations of all pairs of simultaneously recorded neurons were constructed. Crosscorrelations containing fewer than 300 spikes were excluded from the analysis. The bins from −10 to 0 ms and from 10 to 50 ms served to calculate baseline mean and s.d. A peak at short latency in a crosscorrelation was defined as at least one bin between 0.5 and 4 ms that was above 6 s.d. from the baseline mean. A pair of cells was not considered if the baseline was not stable, i.e. if a bin between −10 and 0 ms or between 10 and 50 ms exceeded 75% of the short latency peak.

## Linear track experiment

A second cohort of mice were trained to run on a linear track. The linear track ($80 \times 5.4$ cm) was made of wood and painted gray. The walls along the long axis of the maze were 1 cm high, except for the last 6.5 cm at the two extremities where the walls were 3 cm high. The walls of the short axis of the track were 16 cm high. A food well and an infrared beam were located 2 and 5 cm away from both ends of the track, respectively. Breaking the beam triggered the release of a food pellet at the opposite end of the track. Two gray walls (80 cm long $\times$ 28 cm high) were located 25 cm away from the long edges of the linear track. A single row of 48 LEDs (80 cm long) was attached to one of the side wall. On the opposite wall, 4 rows of 12 LEDs (20 cm long) were attached, aligned to one end of the track. The LEDs were the only potential source of visible light in the recording environment when the mouse ran on the linear track.

After initial training (see above), the mice were trained on the linear track 3 times a day for 10 min. During the first 2 days, food pellets were placed randomly on the track during training. The pellets were progressively moved away from the center of the maze over the next 2 days. Thereafter, pellets were only delivered upon infrared beam breaks. The recording cable was connected to the mouse during the subsequent training sessions (15 min) and training continued until the mouse performed approximately 40–55 runs within 15 min.

Recording sessions with the linear track started with 20 min of exploration in a $70 \times 70$ cm open field with normal light illumination, followed by 20 min in a rest box ($25 \times 25$ cm). The recording session continued with 3 20-min trials on the linear track separated by 20-min trials in the rest box. There were 3 lighting conditions (l1, l2 and d) on the linear track. The order of presentation was random and the condition changed every 5th run on the track. In l1, the single row of LEDs was turned

on. There was no other visible light source in the room. In l2, the 4 rows of LEDs were turned on instead of the single LED row. In d, all LEDs were turned off. Only recording sessions with at least 20 blocks of 5 runs on the track were considered for analysis.

Identification of spatially selective neurons in mice trained on the linear track was performed using the data from the square open field. The 95th percentiles of the surrogate distributions served as significance levels for each spatial score. The border score was defined as $(CM - DM)/(CM + DM)$.

## Linear firing rate maps

The position data on the linear track were linearized by calculating the regression line of the two-dimensional position data. Each position coordinate was moved to the closest point on the regression line. Linear firing rate maps were calculated like two-dimensional firing rate maps (same parameters and smoothing), with the exception that the data were unidimensional and runs toward each end of the track were treated separately.

## Histology

To confirm tetrode location, mice were deeply anesthetized with ketamine and xylazine, and perfused transcardially using saline, followed by 4% paraformaldehyde. The brains were removed and stored in 4% paraformaldehyde at 4°C overnight. The brains were then sliced in 50 μm-thick slices and stained with cresyl violet.

## Acknowledgements

This work was supported by an Emmy Noether Program grant (AL 1730/1-1) to KA and a Collaborative Research Centre (SFB-1134) from the DFG. We thank Prof. Hannah Monyer for many fruitful discussions and comments on the manuscript.

## Additional information

### Funding

| Funder | Grant reference number | Author |
|---|---|---|
| Deutsche Forschungsgemeinschaft | AL 1730/1-1 | Kevin Allen |
| Deutsche Forschungsgemeinschaft | SFB-1134 | Kevin Allen |

The funders had no role in study design, data collection and interpretation, or the decision to submit the work for publication.

### Author contributions

JAP-E, KA, Conception and design, Acquisition of data, Analysis and interpretation of data, Drafting or revising the article; OK, Analysis and interpretation of data, Drafting or revising the article; PL, LK, Acquisition of data, Analysis and interpretation of data, Drafting or revising the article

### Author ORCIDs

Kevin Allen, http://orcid.org/0000-0001-5319-3721

### Ethics

Animal experimentation: All experiments were carried out in accordance with the European Committees Directive (86/609/EEC) and were approved by the Governmental Supervisory Panel on Animal Experiments of Baden-Württemberg at Karlsruhe (35-9185.81/G-113/10). Every effort was made to minimize suffering.

## Additional files

### Major datasets

The following dataset was generated:

| Author(s) | Year | Dataset title | Dataset URL | Database, license, and accessibility information |
|---|---|---|---|---|
| Pérez-Escobar JA, Kornienko O, Latuske P, Kohler L, Allen K | 2016 | Data from: Visual landmarks sharpen grid cell metric and confer context specificity to neurons of the medial entorhinal cortex | http://dx.doi.org/10.5061/dryad.c261c | Available at Dryad Digital Repository under a CC0 Public Domain Dedication |

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
