## [Decision Letter]

Thank you for submitting your article "Visual landmarks sharpen grid cell metric and confer context specificity to neurons of the medial entorhinal cortex" for consideration by *eLife*. Your article has been reviewed by two peer reviewers, including Matthew F Nolan, and the evaluation has been overseen by a Reviewing Editor and Sabine Kastner as the Senior Editor.

The reviewers have discussed the reviews with one another and the Reviewing Editor has drafted this decision to help you prepare a revised submission

Summary:

This study presents interesting and important data demonstrating that firing of grid and other spatial cells in the medial entorhinal cortex (MEC) is dependent on visual landmark information. The study uses elegant experimental designs to reveal striking effects of visual stimuli. The results are exciting first because grid firing has previously been assumed to be stable in the dark – the experiments provide compelling evidence this is not the case – and second as they will very likely stimulate new directions of investigation into mechanisms responsible for grid firing patterns. The reviewers found much merit with the study but also had a large number of specific concerns that can be addressed with additional analyses and explanation.

1) The analysis of grid cells in the dark period could be expanded. Does the grid map remain rigid in the dark period, but drift in its orientation and phase? A previous paper looked at grid cell error accumulation as a function of distance from the boundaries of the environment, and also assessed whether grid fields were drifting by looking at error spikes, i.e. spikes that occurred outside of pre-defined firing fields (Hardcastle et al., 2015). Can the authors perform a similar analysis in their dark period data and report the rate at which error accumulates in their conditions?

2) How much this result is influenced by the regular alternation of the cue location? Is it possible that this reduces the stability of the grid network by regularly introducing large error signals?

3) Different parts of the MEC differ in terms of their anatomical inputs from visual regions. Did the influence of visual cues vary with tetrode depth, or grid scale?

4) Please show histology for all mice. Do the results differ at all if only recording sites within the MEC are kept?

5) Including grid cells with grid score > 0.5 during l1 and l2 trials will artificially inflate the decrease in grid score during dark trials. The authors should exclude these cells unless they also passed threshold during both baseline periods.

6) The thresholds used to identify grid cells appear to be fairly conservative. Regardless, the authors should perform a shuffling analysis showing the expected null distribution of grid, information, CM, DM, and polarity scores in their dataset to justify their choice of thresholds for all classes.

7) What were the criteria for deciding the tetrodes were in MEC to start recording?

8) How were the lights controlled during experiments?

9) How were food rewards delivered during experiments?

10) When identifying putative excitatory connections, were all pairs of simultaneously recorded neurons used, across both hemispheres, or just those on the same tetrode?

11) In the statistical analysis neurons are treated as being independent from one another. This is unlikely to be justified for neurons recorded from the same animal. When comparing light versus dark a more correct approach would be to use linear mixed effect models with animal identity and session included as random effects. The raw data do look pretty convincing but this would help remove any nagging doubts.

On a somewhat related note, it wasn't clear whether data recorded from the same animal in multiple sessions might be from the same neuron(s). Could this be the case? How is this avoided?

12) Do the firing rate changes between light and dark trials differ for putative interneurons compared to principal cells? In Figure 6, cells 1 and 2 have high firing rates suggesting they might be interneurons, but it’s not clear if interneurons are included in Figure 6.

13) Was head direction firing affected by visual landmarks? The MEC contains many neurons with activity modulated by head direction. Head direction was measured, but I could not find effects of visual modulation on head direction firing reported in the main text, although firing rate changes of head direction cells are in a couple of the figures.

14) What fraction of speed cells was modulated by head direction? What information do the cross-correlations reveal about post-synaptic targets of speed cells? Do any post-synaptic neurons show reductions in firing rate as one would expect if speed cells are inhibitory?

---

## [Author Response]

Summary:

*This study presents interesting and important data demonstrating that firing of grid and other spatial cells in the medial entorhinal cortex (MEC) is dependent on visual landmark information. The study uses elegant experimental designs to reveal striking effects of visual stimuli. The results are exciting first because grid firing has previously been assumed to be stable in the dark – the experiments provide compelling evidence this is not the case – and second as they will very likely stimulate new directions of investigation into mechanisms responsible for grid firing patterns. The reviewers found much merit with the study but also had a large number of specific concerns that can be addressed with additional analyses and explanation.*

*1) The analysis of grid cells in the dark period could be expanded. Does the grid map remain rigid in the dark period, but drift in its orientation and phase? A previous paper looked at grid cell error accumulation as a function of distance from the boundaries of the environment, and also assessed whether grid fields were drifting by looking at error spikes, i.e. spikes that occurred outside of pre-defined firing fields (Hardcastle et al., 2015). Can the authors perform a similar analysis in their dark period data and report the rate at which error accumulates in their conditions?*

We have now used the spike distance metric (SDM) developed by Hardcastle and coworkers to estimate the rate of error accumulation in the firing activity of grid cells during dark trials. SDM increases rapidly at the beginning of dark trials. These results are now shown in Figure 2—figure supplement 1.

We have expanded the section on the firing of grid cells in darkness. An analysis of the firing rate associations between pairs of grid cells (correlation of instantaneous firing rate vectors) during light and dark trials was added. The firing associations were highly correlated between l1 and d1 trials (r = 0.926). This indicates that the firing associations of grid cells are largely conserved despite the loss in spatial stability in darkness. This result is now part of Figure 2.

The analysis of spike-triggered firing rate maps demonstrates that the modulation of firing by traveled distance in grid cells is impaired but above chance levels in darkness over periods shorter than 10 s. This suggests that the phase drift during a 10 s time window is sufficient to affect distance coding. We have now added a figure in the manuscript showing that the head-direction preference of head direction cells is strongly reduced during dark trials (Figure 6). Taken together, these results suggest that the grid pattern might still be present but drifts rapidly in phase and orientation in darkness.

*2) How much this result is influenced by the regular alternation of the cue location? Is it possible that this reduces the stability of the grid network by regularly introducing large error signals?*

This is a very interesting question. The instability in darkness was much larger than that observed in Hafting et al. (2005, Nature) or than that in a previous experiment from our group (Allen et al., 2015, Journal of Neuroscience). Many factors are possibly involved. For example, the circular arena in the current experiment had no wall. This likely removed a potential source of polarizing cues from the environment. In addition, using two light cues located at 90 or 180 degrees to each other can have at least 2 effects. As pointed out by the reviewer, it might generate large error signals when the light is switched back on after a dark trial and this might affect how the animal anchors its internal representation to the external world. A second consequence of having 2 lights is that uncontrolled cues left on the arena by the animal do not have a stable position within the internal representation of the animal. It is possible that these local cues normally contribute to the spatial stability of the grid network in darkness and that using two different lights reduces their contribution.

The laboratory of Tom Wills and Francesca Cacucci presented a poster at the iNav 2016 meeting in Austria in which they reported that grid cells periodicity in mice was disrupted in darkness. Their recording protocol did not involve any cue rotation. This suggests that the reduction of grid periodicity in darkness is not only due to the use of 2 alternating lights.

*3) Different parts of the MEC differ in terms of their anatomical inputs from visual regions. Did the influence of visual cues vary with tetrode depth, or grid scale?*

We have tested this by calculating the correlation between grid spacing and the magnitude of the firing rate change between light and dark trials. The correlation was not significant (n = 132 grid cells, r = 0.073, p = 0.407).

It should be noted that the tetrodes were targeted to the most dorsal part of the MEC. It remains possible that the effect of visual landmarks on the firing of grid cells varies with tetrode depth but that recordings from more ventral regions are required to observe the effect.

We did not include the correlation analysis in the manuscript because of this potential caveat. If the reviewers think that this result should be added to the manuscript, we will do so upon request.

*4) Please show histology for all mice. Do the results differ at all if only recording sites within the MEC are kept?*

We have added source data files that contain the histological results of 12 mice. The brain of one mouse trained on the circular arena could not be recovered at the end of the experiment. All brain sections containing tetrode tracks are shown. In addition, we added two tables indicating the tetrode locations in each mouse.

As the four tetrodes implanted in each hemisphere were often very close to each other, we prefer not to assign the recorded signals to specific tetrode tips. Only tetrodes implanted in a hemisphere in which all tetrode tips were in the MEC were labeled as “MEC tetrodes”.

We now report that the differences in grid score, information score, speed score, borderness (DM), map polarity, head-direction vector length, and firing rate between l1 and d1 trials are still observed when considering only neurons recorded on MEC tetrodes.

*5) Including grid cells with grid score > 0.5 during l1 and l2 trials will artificially inflate the decrease in grid score during dark trials. The authors should exclude these cells unless they also passed threshold during both baseline periods.*

We agree with the reviewer that using the data from l1 trials artificially inflated the decrease in grid score during d1 trials. This has been corrected.

While developing the original manuscript, we initially used only the two baseline periods to identify grid cells. However, in some recording sessions the coverage of the circular arena was incomplete during the 10-min baseline periods and prevented grid cells from reaching the detection threshold. We therefore looked for alternative strategies to classify neurons that displayed clear grid periodicity in their l1 and l2 maps as grid cells.

In the revised manuscript, grid cells are defined as neurons with a significant grid score during either the two baseline periods or during one baseline period and l2 trials. Because we only use the data from l1 and d1 trials to assess the effect of visual landmarks, our detection procedure should not affect the magnitude of the differences observed.

If the reviewers think that using only both baseline periods would be more appropriate, we could modify the procedure. This would not affect the conclusions of the manuscript but would significantly reduce the number of neurons classified as grid cells and clear grid cells would be falsely classified as irregular spatially selective cells.

*6) The thresholds used to identify grid cells appear to be fairly conservative. Regardless, the authors should perform a shuffling analysis showing the expected null distribution of grid, information, CM, DM, and polarity scores in their dataset to justify their choice of thresholds for all classes.*

We have performed a shuffling procedure to obtain null distributions for all our spatial measures during the baselines and trials (Figure 1—figure supplement 1). The 95^th^ percentiles are used to identify the different cell classes.

To be member of a functional cell class, a neuron had to reach the thresholds for the class during either the two baseline periods or during one baseline period and l2 trials.

Below are the new definitions of the different cell classes.

Grid cell: grid score > 95^th^ percentile

Border cell: border score > 95^th^ percentile

Irregular spatially selective cell: information score > 95^th^ percentile (not classified as a grid or border cell)

Head-direction cell: head-direction vector length > 95^th^ percentile, head-direction tuning curve peak > 5 Hz.

Speed cell: speed score > 95^th^ percentile.

*7) What were the criteria for deciding the tetrodes were in MEC to start recording?*

The work of Fyhn et al. (2008, Hippocampus) shows that the power of theta oscillations increases significantly when a tetrode reaches the MEC. In our experiment, the tetrodes were lowered over a few days while the unfiltered electrophysiological signals were monitored on an oscilloscope. Recording started when large theta oscillations were observed on most tetrodes. Typically, the spikes of neurons were also strongly modulated by theta oscillations.

This information was added to the Methods section.

*8) How were the lights controlled during experiments?*

An open-source microcontroller (Arduino Uno) controlled the lights via a 4-channel relay module. The identity of l1 and l2 and the order of presentation were determined with the random function of the Arduino software. The initialization of the random sequence was different for each session.

This information was added to the Materials and methods section.

*9) How were food rewards delivered during experiments?*

The food rewards (AIN-76A Rodent 378 tablets 5 mg, TestDiet) were delivered with pellet dispensers (CT-ENV-203-5, MedAssociates) located on the roof of the recording enclosure. Food delivery was triggered by an Arduino Uno with inter-delivery intervals ranging from 20 to 40 s.

This information is now reported in the Materials and methods section.

*10) When identifying putative excitatory connections, were all pairs of simultaneously recorded neurons used, across both hemispheres, or just those on the same tetrode?*

All possible pairs of simultaneously recorded neurons were considered, but putative connections were only detected between neurons recorded on the same tetrode. Below are the numbers of pairs considered.

Total number of pairs considered: 10880

Within tetrode pairs: 3292

Between tetrode pairs: 7588

Within hemisphere pairs: 7328

Between hemisphere pairs: 3552

Total number of connected pairs: 61

Within tetrode connected pairs: 61

Between tetrode connected pairs: 0

Within hemisphere connected pairs: 61

Between hemisphere connected pairs: 0

*11) In the statistical analysis neurons are treated as being independent from one another. This is unlikely to be justified for neurons recorded from the same animal. When comparing light versus dark a more correct approach would be to use linear mixed effect models with animal identity and session included as random effects. The raw data do look pretty convincing but this would help remove any nagging doubts.*

We have considered using linear mixed effect models to account for the dependence of some data points in our experiment. For example, we built a model to predict grid scores based on the factors condition (l1 vs d1), animal and session. The factors animal and session were included as random effects, while condition was treated as a fixed effect. The factor condition was significant (p < 10^-16^). However, the variance in the residuals of l1 and d1 trials was significantly different (Levene's test for homogeneity of variance: df = 1, F = 113.11, p < 10^-16^). Therefore, the model did not fulfill the assumption of homoscedasticity which is required for linear models. For this reason, we would prefer not using linear mixed effect models in the manuscript.

To address the reviewers' concern regarding the dependence of the data points, we confirmed that the main conclusions of the manuscript remained unchanged when using a single data point per mouse. For example, when using the median grid score per mouse as a statistical unit, we observed a significant difference between l1 and d1 trials (Figure 2, paired Wilcoxon signed rank test, n = 6 mice, p = 0.031). There was only one occasion where the p value did not reach significance level (comparison of map polarity in light and dark trials for border cells, p = 0.11).

If the reviewers think that it would be more appropriate to use linear mixed effect models, we will include them upon request. As they have much more statistical power than using mice as statistical units, we do not expect changes in the conclusions.

*On a somewhat related note, it wasn't clear whether data recorded from the same animal in multiple sessions might be from the same neuron(s). Could this be the case? How is this avoided?*

It is possible that on some occasions the data from successive days contain the same neuron twice. We tried to minimize this possibility by lowering our tetrodes by approximately 25-50 μm between recording sessions.

The statistical analysis performed using mice as statistical units should ensure these duplicates do not affect the conclusions drawn in the manuscript.

*12) Do the firing rate changes between light and dark trials differ for putative interneurons compared to principal cells? In Figure 6, cells 1 and 2 have high firing rates suggesting they might be interneurons, but it’s not clear if interneurons are included in Figure 6.*

All neurons were included in Figure 6. We have included an example of a high firing rate neuron in the figure and modified the figure legend to make this clear.

The proportion of neurons with significant rate changes between l1 and d1 trials was similar in the two populations (putative interneurons: 0.617, putative principal cells: 0.549). This finding is now reported in the Results section.

*13) Was head direction firing affected by visual landmarks? The MEC contains many neurons with activity modulated by head direction. Head direction was measured, but I could not find effects of visual modulation on head direction firing reported in the main text, although firing rate changes of head direction cells are in a couple of the figures.*

Head direction selectivity of neurons was severely reduced in darkness. A new figure (Figure 6) now presents this finding.

*14) What fraction of speed cells was modulated by head direction?*

We now report the number of speed-modulated cells in the different functional cell classes.

Grid cells: 67 out of 139

Irregular spatially selective cells: 54 out of 226

Border cells: 11 out of 63

Head-direction cells: 30 out of 85

30 out of 304 speed-modulated cells were modulated by head direction.

*What information do the cross-correlations reveal about post-synaptic targets of speed cells?*

Putative excitatory connections detected from the crosscorrelations usually involve a low firing rate pre-synaptic cell and a high firing post-synaptic interneuron. If we take into consideration connected pairs in which the pre-synaptic neuron was a speed cell, the mean firing rate of the post-synaptic neurons was 19.80 Hz (n = 17). Out of the 17 post-synaptic neurons to speed-modulated cells, 12 were speed-modulated cells, 13 had a mean firing rate above 10 Hz and 3 were grid cells.

We decided not to include this information in the manuscript because it is based on a relatively small sample (n = 17).

*Do any post-synaptic neurons show reductions in firing rate as one would expect if speed cells are inhibitory?*

Inhibition cannot be unequivocally identified from the crosscorrelations because its effect is likely to last several milliseconds. This makes it very difficult to disentangle synaptic inhibition from the effect of network oscillations. For example, consider the crosscorrelation between two theta-modulated cells, where the preferred theta phase of cell A comes 30 degrees after cell B. The crosscorrelation between cell A and cell B will have fewer spikes after time 0, even if there is no synaptic inhibition of cell B by cell A.

For this reason, we did not attempt to quantify inhibition from the crosscorrelations.

In the manuscript, several speed-modulated cells are spatially selective neurons with relatively low firing rate (i.e., grid cells and irregular spatially selective cells). Moreover, in the analysis of putative excitatory connections, some speed-modulated cells were found to be pre-synaptic neurons. This indicates that speed-modulated cells can also be excitatory cells.